# Fermented *Carica papaya* and *Morinda citrifolia* as Perspective Food Supplements for the Treatment of Post-COVID Symptoms: Randomized Placebo-Controlled Clinical Laboratory Study

**DOI:** 10.3390/nu14112203

**Published:** 2022-05-25

**Authors:** Zaira Kharaeva, Albina Shokarova, Zalina Shomakhova, Galina Ibragimova, Pavel Trakhtman, Ilya Trakhtman, Jessie Chung, Wolfgang Mayer, Chiara De Luca, Liudmila Korkina

**Affiliations:** 1Microbiology, Immunology, and Virology Department, Kabardino-Balkar Berbekov’s State University, 360022 Nal’chik, Russia; irafe@yandex.ru; 2COVID Unit, Rehabilitation Centre, State Hospital N1, 360022 Nal’chik, Russia; shokarova2008@mail.ru (A.S.); shomakhovaz@mail.ru (Z.S.); 3Centre for Innovative Biotechnological Investigations Nanolab (CIBI-NANOLAB), 117437 Moscow, Russia; nint2000@yandex.ru; 4Blood Bank, Federal Centre for Paediatric Haematology, Oncology and Immunology, 117437 Moscow, Russia; pavel.trakhtman@fccho-moscow.ru; 5R&D, Swiss DEKOTRA Ltd., CH-8048 Zurich, Switzerland; trakhtman@dekotra.com; 6R&D, Natural Health Farm, Shah Alam 40150, Selangor, Malaysia; jessiechung.nhf@gmail.com; 7R&D, Medena AGAffoltern-am-Albis, CH-8910 Zurich, Switzerland; wolfgang.mayer@medena.ch (W.M.); chiara.deluca@medena.ch (C.D.L.)

**Keywords:** post-COVID symptoms, fermented food supplements, *Carica papaya*, *Morinda citrifolia*, pro-inflammatory cytokines, oral-nasal-pharyngeal microbiota, antioxidant activity, nitrates/nitrites, phagocytosis, ATP

## Abstract

Food supplements based on fermented *Carica papaya* and *Morinda citrifolia*, known for their immune modulating, redox balancing, and anti-inflammatory effects, were added to conventional treatment protocols prescribed to patients recovering after severe and moderate COVID-19 disease in order to alleviate long-lasting post-COVID symptoms. A randomized single-center placebo-controlled clinical laboratory study was designed and performed (total number of participants 188, with delta variant of virus 157, with omicron 31). Clinical statuses were assessed using computer tomography, electrocardiography, a questionnaire, and physical endurance. Plasma cytokines (IL-6, IL-8, IL-17A, and INF-gamma), nitrate/nitrite ratio, antioxidant activity (AOA), and polymorphonuclear leukocyte (PMN) ATP levels were determined before and 20 days following the addition of 28 g of fermented supplements twice per day. The capacity of PMN to phagocyte and the oral-nasal-pharyngeal microbiota were assessed. Clinical symptoms, IL-6, IL-8, and nitric oxide metabolites diminished significantly compared to the placebo group and their background expression. The PMN capacity to phagocyte, AOA, and ATP content remarkably increased. The oral-nasal-pharyngeal microbiota were unchanged. On these grounds, we suggest that fermented tropical fruits could efficiently diminish post-COVID clinical symptoms through several immune-modulating, redox balancing, and pro-energy mechanisms.

## 1. Introduction

The COVID-19 pandemic has affected millions of people around the world and is considered as a major health threat in many countries. This infectious disease caused by severe acute respiratory syndrome coronavirus (SARS-2) affects not only the lungs, but many other vital organs as well, resulting in a high-risk of severe morbidity and mortality [1,2]. COVID-19 has been associated with multiple cardio-vascular complications, such as acute myocardial injury, myocarditis, arrhythmias, and thromboembolism (reviewed in [3]). Due to virus-induced vascular abnormalities, patients could be affected by “silent” hypoxia, which has been detected in more than 50% of patients [4].

Although patients can recover from the viral infection, some of the side-effects may have a significant impact on recovered patients in the future. A variety of symptoms expressed beyond the acute phase of COVID-19 are commonly called post-COVID syndrome. The syndrome could last more than 200 days after the COVID-19 disease was diagnosed. It is characterized by musculoskeletal, pulmonary, digestive, neurological, and cardio-vascular disorders [5]. The most common complaints include weakness, fatigue, headache, dyspnea, attention deficit, hair loss, myalgia, and arthralgia [6].

Notwithstanding the fast-growing search for clinically efficient, safe, and cost-effective means for the management of undesirable post-COVID symptoms, there are still unmet clinical needs and social concerns.

A large body of literature suggests that the implementation of proper nutrition and supplementation can be helpful in the prevention and fighting of respiratory viral infections, including COVID-19. Several vitamins, microelements, and n-3 fatty acids have been considered as a cost-effective strategy for maintaining optimal immune function and reducing the negative impact of viral diseases (reviewed in [7,8,9,10]). 

According to leading views, practically all negative life-threatening outcomes of COVID-19 disease, such as acute respiratory distress syndrome, cardio-vascular damage, and pro-thrombotic conditions, are secondary to the overload of pro-inflammatory cytokines (cytokine storm) and to aggressive oxidative stress [2,10]. As a result, dietary vitamins (A, D, E, and group B) and microelements (zinc, iron, copper, selenium, and magnesium) possessing anti-inflammatory properties, able of inducing anti-oxidative response, inhibiting the production of pro-inflammatory cytokines through various mechanisms, and stimulating innate immunity against infections, have been recommended as a supplement for infection prevention [9,11,12,13] and as remedies for COVID-19 disease and its consequences [9,10,12]. Several other important nutritional factors, such as n-3 fatty acids [14] and probiotics/prebiotics derived from *Lactobacillus* and *Bifidobacteria* [14,15], have been suggested for the prevention and treatment of COVID-19-related gastrointestinal and lung disorders

Historically well-known plant-based remedies in ethnopharmacology and nutrition have regained their importance amid recent phytochemical, molecular, and preclinical findings. The popularity of non-mineral and non-vitamin dietary supplements, such as botanical supplements, has occupied the primary position as an alternative health approach for the past twenty years [16]. Recently, many botanical preparations and dietary plant-based products have been proposed for the prevention and treatment of COVID-19 [17,18,19,20,21,22]. 

Unfortunately, practically all publications on the matter have serious limitations since they are based on the theoretical assumption that these substances have been found to be clinically efficient towards several health problems in the past. The major critical issue is the scarce presence or even lack of reliable clinical data on their efficacy against COVID-19 disease and post-COVID symptoms.

Since the late 1990s, many research teams have been investigating the biological and health-promoting properties of fermented tropical fruits, such as *Carica papaya* L. (commonly known as papaya), *Moringa citrifolia* L. (commonly known as noni), *Garcinia mangostana* (commonly named mangosteen), wild berries, *Ananas comosus* (known as pineapple), and others (reviewed in [16]). These fermented products, being confirmed pre-biotics/pro-biotics [23], are modulators of redox status [16,24,25,26,27] and the immune system [28,29], are natural antibiotics [16,30] and anti-viral [31,32], anti-obesity [27,33], and anti-diabetics [34,35] with anti-inflammatory [29,36,37,38], are regeneration-accelerating [39,40,41], memory-improving [42,43], and stress-regulating [44], and seem to be suitable candidates for applications as health food supplements for people slowly recovering from severe and moderate COVID disease.

Fermentation is the most ancient and most natural way of plant food and folk medicine processing and preservation. During the fermentation (or external digestion) of plant parts—such as high molecular weight molecules including polysaccharides, nucleic acids, lipids, and glycolipids/glycoprotein—are oxidized by fermenting the enzymes of bacteria, mold, or yeasts, and digested up to low-molecular weight units (molecular moieties), which also occurs in the human gut. These small molecules are readily absorbed and immediately available for a variety of metabolic processes essential to maintaining the organism’s structure and function. This natural way of fruit processing releases mineral micro-elements from their phytate complexes. Highly bio-active fermented fruits are additionally enriched with the membrane fragments and lipo-glyco-peptide complexes derived from the fermenting yeasts and lactobacilli, which contribute additional pre/pro-biotic, nutritional, physiological, and pharmacological values.

Noni and papaya fruits have been used in the traditional medicines of tropical/subtropical South East Asia over more than 1000 years. It has been attracting an ever-growing interest in Western medicine lately and is widely sold in the form of pills, tablets, and fermented powders, purees, syrups, and juices [45]. Fermented noni supplements are of particular importance for their multiple and pronounced health properties ([33,34], see references above). Based on a toxicological assessment, fermented noni juice/puree was considered as safe [46]. The noni fruit is inedible due to its unpleasant smell and unpalatability amid a large percentage of volatile butenyl esters, sulfur-containing volatiles, and acids such as hexanoic, octanoic, and butylic acid [47]. The esters and acids seem to be produced in the process of nonioside (a unique glycoside in the noni fruit) decomposition [48]. In general, approximately 200 biologically active substances were identified and some of them isolated from fermented noni fruits [16]. The content of phenolic acids, flavanols, and polysaccharides are greatly changed during the long-lasting fermentation process. As a result, fermentation led to much higher flavanol, phenolic acids, and lower polysaccharide levels [47]. Easily available polyphenols and anthocyanides determine the remarkably increased antioxidant potency of fermented papaya and noni compared to fresh juices/purees [49]. Several compounds with anti-inflammatory activity, such as glycosylated iridoid asperulosidic acid, rutin, nonioside, glucopyranoside, and tricetin, have been identified in fermented noni juice [36]. They inhibited inducible NO-synthase and down-regulated the expression of Iκκα/β, I-κBα, and NF-κB p65 in the pro-inflammatory cells. The anti-inflammatory effects of orally administered fermented noni were associated with the induction of quinone reductase-1 [37] and the down-regulation of pro-inflammatory cytokines, such as IL-1β, IL-4, IL-6, IL-8, and TNF-α. In contrast, enhanced immunity induced by fermented noni against parasites, microbes, and viruses was connected to the activation of macrophages [31,32], to the up-regulation of INF-γ, and the down-regulation of IL-4 [28].

Fermented papaya preparations have been used in the folk medicine of South East Asia throughout history. They are considered in the Asian pharmacopoeias as remedies for sore throats and as wound healing, anti-malarial, anti-bacterial, abortive, and purgative medicines. They have become extremely popular in Western medicine in the wake of numerous phytochemical, pharmacological, and clinical research and discoveries of the mechanisms underlying its health properties. Now fermented papaya products are commercialised around the world. Fermented papaya has anti-inflammatory, antioxidant, and immune-modulatory properties [50,51,52,53] and has been shown to have anti-diabetic effects in clinical studies [54,55]. The fermented papaya preparation enhanced compromised intracellular bacterial killing from monocytes in diabetic patients by inducing NADPH-oxidase, a key “oxidative burst” enzyme fighting bacterial infections [56]. The intracellular bacteria killing was augmented by the topical application of fermented papaya gel to oral epithelia and the suppression of bacterial catalase, an inducible enzyme protecting bacteria from the oxidative burst of the host phagocytes [52]. The pronounced redox balancing properties of, and the stimulation of phagocytosis by fermented papaya, have been demonstrated [51,57,58].

Keeping in mind that causative reasons for adverse post-COVID complications, such as a greatly altered immune response shifting from over-reaction (cytokine storm, increased immunoglobulins, and acute generalised inflammation) to immune suppression (lymphopenia, secondary bacterial/viral infections, and chronic inflammation) and peculiar redox changes could be balanced by the fermented papaya and noni (see references above), two widely commercialised fruit syrups at affordable costs. As such, they might be a clinically efficient, safe, and cost-effective means of suppressing post-COVID health symptoms.

Here, we attempted to obtain clinical proofs of efficacy for two fermented food supplements (FFS) in the form of syrups based on *Carica papaya* and *Morinda citrifolia* in a randomized and placebo-controlled clinical-laboratory study focusing on long-lasting post-COVID symptoms. Also, the possible mechanisms of FFS action were investigated by assessing anti-bacterial immunity, anti-inflammatory pathways, redox balance, and energy production/expenditure.

## 2. Materials and Methods

### 2.1. Patients and Study Design

The diagnosis of COVID-19 infection was confirmed by a specific polymerase chain reaction (PCR) for the presence of SARS-CoV-2 RNA in the oropharyngeal-nasopharyngeal swabs described elsewhere. The study protocol was scrutinized and approved by the local Ethical Committee (EC of Berbekov’s State Medical University, Nal’chik, Russia; Protocol of the 23d of November 2021). Informed consent was obtained from all participants prior to their enrolment and data collection. Collected data included: personal and anamnestic records, blood sampling for the specific sets of analyses, and blood fractions banking in accord with the Helsinki Declaration on ethics in human experimentations.

The tested cohort consisted of 188 adults (of both sexes aged from 36 to 69 years), who suffered from COVID disease induced by delta (n = 157) and omicron (n = 31) variants of SARS-2-COVID-19 virus and were randomly placed into four groups. Practically healthy donors (n = 25) matching in age and sex were asked to donate venous blood (20 mL) after their informed consent was obtained and the goals of the study were explained. Demographic data of all groups are collected in Table 1. Donors with overt endocrine pathology, acute illnesses, heart diseases, uncontrolled hypertension, current use of hypnotics, or any treatment for breathing disorders, were excluded from enrolment. The disease diagnosis, assessment of clinical conditions, and routine haematological or biochemical analyses of patients during COVID-19 are collected in Appendix A.

### 2.2. Clinical Conditions and Treatment in the Post-COVID Period

At the time of discharge from the hospital, the resolution of fever and an oxyhemoglobin saturation (SpO_2_) ≥94% at rest were required. Post-COVID patients from Group 1 (n = 64) and Group 2 (n = 27) with long-lasting symptoms of adverse effects after severe COVID disease were admitted to the specialized Rehabilitation Centre for three to four weeks. Thirty patients from Group 1 and nine patients from Group 2 remained on respiratory support from oxygen supplementation at the time of discharge. The clinical conditions were routinely assessed by clinicians specialized in rehabilitation. The KT, ECG, and physicality parameters were assessed weekly. Biochemical and hematological analyses were carried out before and after the clinical study. All recruited post-COVID patients were prescribed vitamin B, blood vessel-relaxing and oxygen flow-improving drugs, anticoagulants, and glucose-controlling drugs.

Post-COVID patients from Group 3 (n = 68) and Group 4 (n = 29), after moderate COVID disease with complaints concerning their health impairment, received regular clinical observations and treatments in ambulatories. Their KT, ECG, and physicality parameters were assessed weekly. Biochemical and hematological analyses were carried out before and after the clinical study. All recruited post-COVID patients were prescribed vitamin B, blood vessel-relaxing and oxygen flow-improving drugs, anticoagulants, and glucose-controlling drugs. 

In general, standard rehabilitation procedures were directed to improve lung functions, such as ventilation, gas exchange, bronchial drainage, to increase blood and lymph flow in the damaged lung tissue, to accelerate oedema resolution, and to suppress inflammation and fibrosis. Also, exercises to increase patients’ physical performance/endurance and to correct muscle weakness were recommended and applied to all participants. The psycho-therapeutic approach was also of great importance in alleviating stress, anxiety, depression, and sleep disorders.

### 2.3. Food Supplements under Investigation and Protocols of the Application

Standardised fermented tropical fruits papaya (*Carica papaya* Linn.) and noni (*Morinda citrifolia* Linn.) syrups were investigated.

To increase the palatability of FFS, whole noni fruits and the peels and seeds of unripe papaya fruits were thoroughly mashed and subjected to prolonged (up to 6 months) fermentation; as a result, the volatile butyl esters disappeared. Then, 5% of honey was added to a final sterilized fermented syrup, after which it acquired a sweet, acidic, and slightly bitter and unpleasant taste. The FFS trade names are BioRex Papaya and BioRex Noni, manufactured by Carica Ltd., in Manila, The Philippines. These FFS are widely marketed and distributed locally and around the world.

Once informed consent was obtained, patients in the treatment groups (Group 1 and Group 3) received FFS, a 14 g (a plastic spoon) BioRex Noni, in the morning after breakfast and 14 g of BioRex Papaya (a plastic spoon) after supper. Patients of Groups 2 and 4 received 5% honey diluted by tap water following the same protocol. Thus, daily carbohydrate consumption in addition to habitual dietary carbohydrates did not exceed the amount of 0.9 g/day. In accordance with previously published clinical data on the supplementation of powder forms of fermented papaya containing 90% glucose, maltose, and fructose (9 g/day for 15 days) ([47] Das), the supplementation did not influence the HbA1c, glucose, and total lipid profile.

### 2.4. Biological Material Collection and Processing

Peripheral venous blood (20 mL) was drawn after overnight fasting into vacuum tubes with ethylene diamine tetra acetic disodium salt (EDTA) as the anticoagulant. Both patient and donor samples were processed and analysed in parallel. The circulating polymorphonuclear leukocytes (PMN) were obtained by double-density gradient centrifugation of 15 mL of whole blood (Histopaque, d = 1.077 and 1.199 g/mL). PMN from the interface were re-suspended in phosphate buffer saline, centrifuged at 1650 rpm for 10 min, and then aliquoted at 5 × 10^6^/vial. Freshly isolated PMN were used in phagocytosis assays. Samples of whole blood (5 mL) were allowed to sediment for 40 min at room temperature; then, plasma supernatant was collected, aliquoted, and stored at −80 °C until quantitative analyses of nitrites/nitrates and cytokines were performed [59].

### 2.5. Haematological and Biochemical Analyses

Routine laboratory analyses included: total blood count (erythrocytes, Hb, leukocytes, and platelets), differential count of leukocytes, urea, creatinine, electrolytes, glucose, alanine amino transferase, aspartate amino transferase, bilirubin, albumin, troponin, ferritin, C-reactive protein, pro-calcitonin, coagulation factors, D-dimer, and interleukin 6 (IL-6).

### 2.6. Reagents and Assay Kits for Laboratory Study

The majority of chemical reagents, ELISA kits for interleukins and interferon measurements, mediums, solvents, and luciferin-luciferase for ATP assay were from Sigma Chemical Co. (St. Louis, MO, USA). The Griess reagent for nitrites/nitrates determination were from Cayman Chem. Co. (Ann Arbor, MI, USA). The manufacturers of other reagents are mentioned within the respective methods.

### 2.7. Redox and Oxidation Marker Assays

Complete differential blood cell counts and metabolic analyses were performed on fresh ethylene diamine tetra-acetic acid (EDTA)-anti-coagulated venous blood of 12 h-fasting subjects. Biochemical assays were performed on peripheral blood plasma, red blood cells (RBC), or total white blood cells (WBC) either immediately (ATP), or within 72 hrs, on sample aliquots stored at −80 °C under argon. The plasma levels of nitrites/nitrates (NO_2_¯/NO_3_¯, expressed as μmoles/L) were measured spectrophotometrically by Griess reagent [60]. The protein content was measured by Bradford method [61] using a micro-plate assay kit (Bio-Rad, Hercules, CA, USA).

### 2.8. ATP Assay

One hundred μl of PMN pellet was stored on ice until analysis. Ice-cold water (990 μL) was added to10 μl of cell pellet, mixed and the lysed cells were kept on ice. The principle of ATP assay is based on the quantitative bioluminescent determination of adenosine 5′-12 triphosphate (ATP), assessed by the Bioluminescence Assay Kit. In the assay, ATP is consumed when firefly luciferase catalyses the oxidation of D-luciferin to adenyl-luciferin which, in the presence of oxygen, is converted to oxyluciferin with light emission. This second reaction is essentially irreversible. When ATP is the limiting reagent, the light emitted is proportional to the ATP present. The measurements of luciferin-luciferase chemiluminescence were performed on a Victor2 1420 multi-label counter, equipped with Wallac 1420 Software (Perkin Elmer, Waltham, MA, USA). Results were expressed as mmol/L (mM). [62].

### 2.9. Immunological Assays

The capacity of circulating PMN to phagocyte bacteria was determined by the following methods: phagocytosis index (number of bacteria engulfed by a single phagocyte), phagocyte number (number of active phagocytes), and intracellular killing of engulfed bacteria [53]. Briefly, 1 mL of PMN suspension (10^6^ cells/mL) was mixed with 1 mL of bacterial suspension (10^7^ cells/mL). The mixture was incubated under continuous shaking at 37 °C for 30 min. Then, smears were prepared on microscopic slides, fixed, and stained by Romanovsky–Giemsa dye. The smears were examined under a microscope, and the percentage of phagocyting-PMN was determined. The remaining mixture was used to assess the intracellular bacterial killing. After centrifugation at 1500× *g* for 10 min, bacterial sediments were collected and diluted to an OD600 of 0.1 with fresh medium, spread onto Petri dishes with an appropriate agar-containing medium, and were allowed to grow at 37 °C for 24 h. The bacteria survival rates were calculated as the colony-forming-units (CFU) of cells co-incubated with granulocytes. The results were expressed as a percentage [63].

The plasma levels of interleukin 6 (IL-6), interleukin 8 (IL-8), interleukin 17A (IL-17A), and interferon gamma (IFN-γ) were measured by the enzyme-linked immunosorbent assay (ELISA) using appropriate antibodies purchased from R&D Systems (Minneapolis, MN, USA), following the manufacturer’s instructions. Cytokine concentrations were expressed in pg/mL of plasma. Each protein was quantified in the linear range of its calibration curve [63].

### 2.10. Oral-Nasal-Pharyngeal Microbiota Determination

The total amount of anaerobic microbes on the oral-nasal-pharyngeal epithelia was determined by the routine microbiological test of swabs by sectoral seeding described elsewhere [64]. The isolation and identification of bacteria was performed using microbiological processes and microscopic observation and confirmed by mass spectrometry MALDI-TOF (Microflex, Brucker, Peoria, IL, USA) [65].

### 2.11. Questionnaire and Physicality Tests

The questionnaire containing 22 questions on patients’ subjective opinions was recommended to be filled out in the beginning and at the cessation of the trial. The answers were scored at 0, 1, and 2. The average scores for the Groups and for the time of answers were calculated as well as an average score of all answers for a Group. The data were presented as the mean ± S.D. 

An assessment of the degree of silent hypoxia and subjective dyspnea elicited by a 6-minute walking test (6MWT) [4] along a 30 m-long hospital corridor was carried out twice: prior to beginning of the trial and at its cessation. Patients were instructed to walk at a self-determined pace for the greatest distance possible for 6 minutes. If needed, they were allowed to pause for a little rest. The heart rate (beats/min), SpO_2_, respiratory rate (breaths/min), and dyspnea were scored from 1 to 10 according to the Borg’s scale [66], and were recorded before and immediately after the 6MWT. Patients from Group 1 and Group 2 still in the need of oxygen supplementation were excluded from the 6MWT.

### 2.12. Statistical Analysis

The statistical analysis of the clinical data was carried out using WINSTAT programs for personal computers (Statistics for Windows 2007, Microsoft, MA, USA). All biochemical and microbiological measurements were done in triplicate and data were statistically evaluated. The reported values were treated as continuous. The normality of data was confirmed using the Shapiro–Wilk’s test. Since the distribution of the data was significantly different from normal, nonparametric statistics were used. The results were expressed as the mean ± SD. 

Mann–Whitney U test was employed for comparison between independent groups of data. 

To evaluate the difference between connected data, the two-tailed Student’s *t*-test was applied and *p* < 0.05 were considered significant.

Correlations were performed using Spearman’s non-parametric coefficient (r_s_). The determination coefficient was obtained according to Chaddock’s scale correlation, where r_s_ 0.7–0.9 was considered high connection, 0.5–0.7—salient connection, and 0.3–0.5—moderate connection. 

Fisher’s exact test was used to determine the significance of the differences between the beginning and cessation of the trial while assessing the scores of subjective opinions of patients on the effects of supplementation. For these tests, *p* < 0.05 was considered significant and 0.05 < *p* < 0.1 a trend towards significance.

## 3. Results

### 3.1. Effects of Fermented Fruits Supplementation on Self-Assessment, Cardiac Functions, and Physical Endurance of Post-COVID Patients

#### 3.1.1. Comparable Effects of Fermented Fruit Supplementation and Placebo on Self-Assessment of Clinical Symptoms of Post-COVID Patients

A special questionnaire for the self-assessment of post-COVID patients was developed and recommended to be filled up by all participants of the trial. The questions were formulated on the basis of major complaints of patients who survived severe COVID disease or suffered from the moderate form of the disease. The questionnaire contained 22 questions to assess the intensity, frequency, or existence of a health problem. The answers were scored from 0 to 2 depending on either the intensity or frequency of a symptom, while existences or absences of a symptom were scored as 0 and 1, respectively. Average scores were calculated for the Groups and a single symptom. The data of self-assessment for post-COVID Groups 1 and 2 (severe COVID disease) before and after the trial are collected in Table 2 and those for Groups 3 and 4 (moderate COVID disease) are shown in Table 3.

For the background of patients from Groups 1 and 2 background (30–40 days after discharge from the hospital and before the trial), average scores for symptoms, such as weakness, impairment of physical and mental working capacities, memory and concentration of attention impairment, and dryness of skin/epithelia, were at a maximum and equal to 2.0 ± 0.0. For the other symptoms, average scores were lower ranging from 1.4 to 0.6 (Table 2). Data collected in Table 2 shows that supplementation with FFS (Group 1) led to a highly significant decrease of the symptom scores for weakness and impairment of physical and mental working capacities compared to both the background state and post-trial results in patients from placebo group (Group 2). Therefore, the significance assessed by a Fisher’s exact test were more than 95% (*p* < 0.05). For two other highly expressed symptoms, such as memory and attention concentration impairment, the difference between FFS and placebo groups could be considered as a trend since P lays in the range of 0.05 and 0.1. For skin and epithelia dryness, there was no difference between the experimental and placebo groups (*p* > 0.05).

Regarding other less pronounced symptoms before the trial (score range 1.4–0.6), statistically significant differences between FFS and placebo groups were obtained for 11 symptoms (see Table 2), while only a trending difference was found for headache and dizziness, and no difference was seen for the change of bodyweight and hair loss. 

The total average scores of all symptoms together were significantly different for he FFS and placebo groups (*p* < 0.05) showing clinical efficacy of FFS.

The subjective opinions of the patients in post-COVID period after moderate COVID disease (Groups 3 and 4) are collected in Table 3**.** All the symptoms had much lower average scores ranging from 1.3 to 0.2 at the beginning of the study compared to patients after severe COVID disease (Table 2). However, again, a statistical significance between FFS and placebo groups was present for weakness and the impairment of physical and mental working capacity. In addition, remarkable improvement and differences between the groups after the trial were found for 5 other symptoms, such as heart pain, arrhythmia, and tachycardia, muscle pain, and skin and epithelia dryness. A trending difference between the groups after the trial was observed for 4 symptoms. The others (10 symptoms, among which were changes of bodyweight and hair loss) were similar for both groups (*p* > 0.05).

Total average scores of all symptoms together were significantly different for FFS and placebo groups (*p* > 0.05) demonstrating clinical efficacy of FFS.

#### 3.1.2. Comparable Effects of Fermented Fruit Supplementation (FFS) and Placebo on Electrocardiography of Patients in Post-COVID Period

Several parameters of cardiac function were assessed by electrocardiography, e.g. dysmetabolic changes in the myocardium and cardiac arrhythmias (partial blockade of the left or right bundle of Hiss, bradycardia, and supraventricular extrasystole) before and after the trial (Table 4 and Appendix B, Figure A1). Importantly, all patients of Groups 1 and 2 recovering after severe COVID exhibited metabolic disorders before the study, while only slightly more than 50% of the patients from Groups 3 and 4 had electrocardiographic features of dysmetabolism in the myocardium. The percentage of patients with metabolic disorders decreased more in the groups (Group 1 and 3) treated with FFS than in the control placebo groups. The decrease was more evident for Group 3 after moderate form of COVID. Similar results were obtained for all three parameters of cardiac arrhythmias. 

#### 3.1.3. Comparable Effects of Fermented Fruit Supplementation and Placebo on Subjective Dyspnea and Physical Load of Patients in Post-COVID Period

Respiratory support by oxygen supplementation was still needed for several patients from Groups 1 and 2 at the 30th–40th days after severe COVID (Table 5). At the trial cessation, per cent of patients on respiratory support decreased for the both groups, however, supplementation with FFS helped more post-COVID patients to stop using oxygen. For these groups, subjective dyspnea assessed by a Borg’s score was equally diminished for both the experimental and the placebo groups. At the same time, Borg’s score was statistically significantly improved as compare to the beginning of the trial in Group 3 and in Group 3 versus Group 4 at the study cessation. Subjective dyspnea remained unchanged before and after the trial in placebo Group 4. Physical load given as a 6 minute-long walking was applied exclusively to post-COVID patients (severe COVID groups) who have no need of oxygen support. This test of tolerance to moderate physical load revealed that oxygen desaturation > than 4% occurred in 53.3% of patients supplemented with FFS versus 71.4% of patients receiving placebo. For technical reasons this test was not applied to post-COVID patients after moderate COVID.

### 3.2. Effects of FFS versus Placebo on Cellular Immunity (Functions of Circulating Phagocytes) and Oral-Nasal-Pharyngeal Microbiota

Figure 1 shows that capacity of circulating PMN to phagocyte (number of engulfed bacteria per phagocyte and per cent of active phagocytes) and kill engulfed bacteria was significantly suppressed 30–40 days after hospital discharge in the post-COVID groups of patients survived severe forms of COVID. For post-COVID patients recovering after moderate COVID, suppression of phagocytes was slight often at the lower margin of normal values range. At the cessation of the 3-week-long study, all the evaluated parameters demonstrated activation of phagocytosis in experimental as well as in placebo groups. However, the stimulation upon action of FFS was statistically more pronounced comparing to placebo.

To evaluate possible mechanisms underlying remarkable activating effects of FFS towards PMN, content of ATP in PMN before and after the trial was determined in groups on FFS supplementation (Figure 2). The figure shows that circulating PMN of patients survived severe COVID had extremely low levels of ATP, which was increased up to normality after FFS supplementation. PMN of patients after moderate COVID had normal levels of ATP before the trial and its content has a trend for increase after the course of FFS although remaining within normality values.

Stimulation of phagocytosis and intracellular bacterial killing by FFS theoretically could affect microbiota presence and spectrum on the surface of oral-nasal-pharyngeal epithelia. Therefore, a total number of facultative anaerobic bacteria on these epithelia and the presence of several definite pathologic strains were determined before and after the trial in the patients of experimental versus placebo groups. Data are collected in Appendix C, Table A1. Total amount of microbes on the above epithelia was initially significantly greater-than-normal in post-COVID patients after severe infection (Groups 1 and 2) while there was only a trend to increased amount of epithelial microbes in post-COVID period of patients after moderate COVID disease (Groups 3 and 4). The total amount of microbes as well as the presence of potentially pathogenic microbes, such as *Staphylococcus aureus*, *Klebsiella pneumoniae*, and *Candida albicans* did not change after supplementation period in all experimental and placebo groups.

### 3.3. Effects of FFS and Placebo on Plasma Levels of Cytokines/Chemokines Regulating Immune Response

The plasma levels of interleukins, which are considered markers of COVID infection/disease were measured before and after the clinical study. The content of IL-6 at the peak of disease/admission to the hospital was also taken into consideration. The results displayed in Figure 3 clearly show that the content of pro-inflammatory cytokine IL-6 was tremendously increased above the normal range of values at the moment of patients’ admission to the hospital, particularly for patients with severe form of the COVID disease (Figure 3a). By the 30th–40th day after discharge from the hospital, plasma levels of IL-6 protein decreased substantially in all groups of post-COVID patients but remained significantly higher-than-normal values (Figure 3b). A three-week-long course of FFS supplementation led to normalisation of IL-6 for Groups 1 and 3 (post-COVID after severe and moderate COVID, respectively). At the same time, in placebo groups (Groups 2 and 4), levels of IL-6 were statistically significantly increased as compare to the experimental groups. The same pattern of changes was observed for plasma levels of chemokine IL-8 (Figure 3c) while no significant difference was observed for the levels of IL-17A and IFNγ (Figure 3d,e).

### 3.4. Effects of FFS and Placebo on the Redox Balance in Plasma of Post-COVID Patients

The oxidative-reductive status (redox balance) in plasma was assessed by the levels of nitrites/nitrates, highly oxidising agents formed in the reaction of nitric oxide (NO) and superoxide anion-radical (O_2_^−.^) and by its total antioxidant activity (AOA), a capacity to inhibit lipid peroxidation. Before the trial, the content of plasma nitrites/nitrates was significantly elevated above the normal values in the groups of patients-survivors of severe COVID and only slightly increased in the groups after moderate COVID (Figure 4a). After FFS course, the levels of these dangerous oxidising agents went down to normality. At the same time, in the placebo groups nitrites/nitrates remained practically unchanged. 

As expected, AOA of plasma was lower-than-normal in post-COVID groups after severe COVID (Figure 4b) and was within the range of normality for post-COVID patients after moderate COVID. The FFS supplementation statistically significantly increased AOA in the Group 1 patients as compare to the background situation and to placebo Group 2. Either FFS or placebo did not affect AOA in the groups of post-moderate COVID.

## 4. Discussion

In the present study, we confirmed previous data [5] that a significant number of patients, especially after severe COVID-19 disease, present a vast clinical spectrum of symptoms during long recovery period (30–40 days), negatively affecting the quality of life, and requiring intense and multidisciplinary rehabilitation protocols. The assessment of clinical pattern was carried out subjectively, on the basis of patients’ opinion expressed as answers to the Questionnaire, where major complaints of post-COVID period were collected [5], and by the objective instrumental methods using ECG and effects of moderate physical load on blood oxygenation and ability to complete the 6-min walking task. The Questionnaire showed that background post-COVID health conditions were strikingly different for survivors from severe COVID-19 and for patients recovering from a moderate form of the disease. Supplementation with FFS resulted in significant improvement of clinical symptoms for all patients (Table 2 and Table 3) superior to that of the placebo groups. Perception of changes in memory and attention impairment as well as anxiety/depression, and headache had a trend to statistical difference between FFS and placebo groups. There were no effects on two common symptoms, namely, body weight change and hair loss. The reason could be too short period of supplementation to achieve visible results.

Collectively, for all 22 clinical symptoms analysed, FFS was remarkably more effective than placebo. 

Subjective opinions on the positive effects of FFS on heart functions (heart pain, arrhythmia, and tachycardia) were supported by the dynamics of ECG data (Table 4 and Appendix B, Figure A1): number of patients with metabolic disorders and arrhythmias diminished substantially versus the beginning of the trial and versus placebo group. These changes were more pronounced in the post moderate-COVID group.

Regarding lung functions (Table 5), the number of post-COVID patients after severe disease requiring oxygen supplementation decreased significantly after FFS course. Majority of patients of this group without the need of respiratory support were able of performing a moderate physical load without interruption due to low blood oxygenation. For the parameter of subjective perception of dyspnea assessed by a Borg’s method, we did not observed difference between the experimental and placebo groups.

The causes and mechanisms underlying post-COVID syndrome are still to be investigated. Several current hypotheses include chronic self-maintaining inflammation, immune system and redox imbalance, an autoimmune phenomenon, and a hormonal component [67,68]. In the present clinical study we attempted to evaluate the effects of FFS on some of these mechanisms: functions of circulating phagocytes PMN crucial for the defence from secondary infections and a source of self-maintaining inflammation. As expected, capacity of circulating PMN to phagocyte bacteria and kill them intracellularly was highly suppressed in the groups of post severe COVID patients (Figure 1). Not surprisingly, FFS restore this functions up to normality. FFS effects were much more evident than those of placebo. Our data correspond to previous publications that fermented papaya acts as a stimulator of macrophage phagocytosis [51,57] and microbial killing through several mechanisms, such as induction of NADPH oxidase [55], oxidative burst [54,56], and suppression of bacterial catalase [53]. At the same time, we failed to find any effects of FFS on the quantity and quality/spectrum of microbes on oral-nasal-pharyngeal epithelia (Appendix C, Table A1). Our previous paper [53] have clearly shown that topical application of fermented papaya gel to gums remarkably protected against microbial infection and inflammation of patients with gingivitis and periodontitis. The discrepancy could be partly explained by the fact that the local administration and long-lasting presence of fermented papaya preparation in the oral cavity could facilitate anti-microbial and anti-inflammatory action of FFP.

For the first time, we observed that a lack of PMN functions may depend on the ATP deficiency in the cells (Figure 2) of patients-survivors after severe COVID-19 and not of patients recovering after moderate disease. By unknown as yet mechanisms, ATP levels in PMN returned to the normal range upon the effects of FFS but not placebo. Energy conserving/generating effect of FFS could be explained in terms of hormesis-like action of fermented fruits, due to which cell functions and their defence against excessive oxidation improve through Nrf2-related mechanisms [40,41,63]. Alkyl catechols are natural redox-sensitive co-factors for Nrf2 activation by moderate oxidative stress. They are available from numerous sources especially arising from biotransformation of common plant compounds by lactobacilli expressing phenolic acid decarboxylase. Thus, *Lactobacillus plantarum*, *Lactobacillus brevis*, and *Lactobacillus collinoides*, which are abundant in traditionally fermented foods and beverages, convert common phenolic acids found in fruits and vegetables to 4-vinylcatechol and/or 4-ethylcatechol [40]. These small molecular weight Nrf2 co-factors have been identified in the fermented noni juice [41].

Protection against oxidative stress in COVID-19 and post-COVID patients is essential because imbalance of normal redox status to the oxidative millie seems to be one of the major causes of post-COVID syndrome [5,67,68]. According to our data, overproduction of highly oxidising agents nitrites/nitrates and their presence in plasma is a characteristic feature of post-severe-COVID period (Figure 4). These potentially dangerous agents were effectively diminished by FFS and not by placebo. Probably, FFS being potent superoxide anion-radical scavenger [51,53,57] prevents O_2_^−.^ from reacting with nitric oxide (NO), thus, decreasing levels of toxic nitrites/nitrates while increasing NO availability for essential functions. The FFS, seems to be not very efficient direct antioxidant as revealed by its scarce effect to the total plasma AOA, however, it could be a perfect indirect antioxidant able to inducing endogenous antioxidant defence mechanisms via Nrf2-mediated pathways.

According to [3], numerous, sometimes, life-threatening COVID complications in cardio-vascular system may depend on the impaired immune response to viral infection and on the therapy implicated in the treatment of acute COVID disease. The presence of IgG, IgA and IgM specific to SARS-CoV-2 proteins did not correlate with the severity and duration of post-COVID syndrome [5]. However, plasma levels of IL-6, IFN-γ, and IL-17 proteins were significantly higher in 2-weeks of post-COVID recovery patients than in healthy controls [69]. It has been assumed that cytokines IL-6, IL-17A, and TNF-α are the major players in the cytokine storm and biomarkers for acute respiratory distress syndrome and mortality in patients with COVID-19 and definite risk factors, for example, obesity [70]. The typical laboratory alterations associated with the disease severity were lymphopenia and extremely high plasma levels of inflammatory cytokines in severe as compare to moderate cases [71,72]. Han et al. [73] have reported that exclusively IL-6 was predictive for severe COVID-19 disease. In post-COVID patients (2 weeks of recovery), content of plasma cytokines (IL-6, INF-γ, IL-17, and TNF-α) decreased but still was significantly superior to normal levels [69]. A meta-analysis of latest literature [72] has shown that biomarkers of inflammation, cardiac and muscle injury, and serum levels of IL-6 were highly increased in patients with severe/fatal form of COVID as compare to non-severe cases.

The published data correlate with our findings on the plasma levels of IL-6, IL-8, and IL-17A exclusively for the groups after severe COVID-19 disease (Figure 3, before the trial). The most predictive biomarker for COVID-19 complications IL-6 was found 10 times decreased by the beginning of the trial (at the 30th–40th days after discharge from the hospital) as compare to its plasma content at the admission to the hospital (Figure 3a,b). As expected, at the entrance to the trial, plasma IL-6 was much higher in post severe COVID versus post non-severe COVID. Still, IL-6 levels remained much higher-than-normal. FFS statistically significantly suppressed plasma IL-6 down to normality. Placebo was slightly effective. The same pattern was observed for the chemokine IL-8 (Figure 3c). Only a trend to difference between the FFS and placebo groups was found for IL-17A (Figure 3d), and no difference for INF-γ (Figure 3e). Effects of fermented fruit supplements on inflammatory cytokines mainly depend on the model of inflammation. Fermented tropical fruit-containing beverages stimulated immune response in the in vitro experiments with isolated cells [51,57], in non-specific inflammatory in vivo model as visualised by enhanced macrophage activity and pro-inflammatory cytokine (IL-6, TNFα, and IFNγ) expression, while these beverages suppressed specific immune response (decreased IL-6 and IL-4 production but increased IFNγ) in the animal model of allergy [50]. Keeping this in mind, suppressive action of FFS to several pro-inflammatory cytokines/chemokines might reflect on its effects to non-specific over-expressed immune response to viral and secondary bacterial infections or to specific auto-immune responses to modified proteins. More studies on the mechanisms of immunomodulation related to FFS are needed.

## 5. Conclusions

Fermented tropical fruits (*Carica papaya* L. and *Morinda citrifolia* L.) supplementation of patients with post-COVID syndrome (after severe and moderate forms of COVID-19 disease) is clinically effective improving deteriorated heart and lung functions as well as positively affecting perception of numerous health problems induced by pre-existing COVID-19 disease. Possible underlying mechanisms include the restoration of compromised immune cell functions and energy storage/generation, the suppression of interleukin-dependent pro-inflammatory millie in the blood, and the normalisation of healthy redox status. This is one of the very first clinical case- and placebo-controlled studies showing clear-cut clinical and metabolic improvements of post-COVID symptoms induced by the consumption of health food supplements. This approach could be considered as safe and cost-effective way to address post-COVID patients’ quality of life and possible future complications

## Figures and Tables

**Figure 1 nutrients-14-02203-f001:**
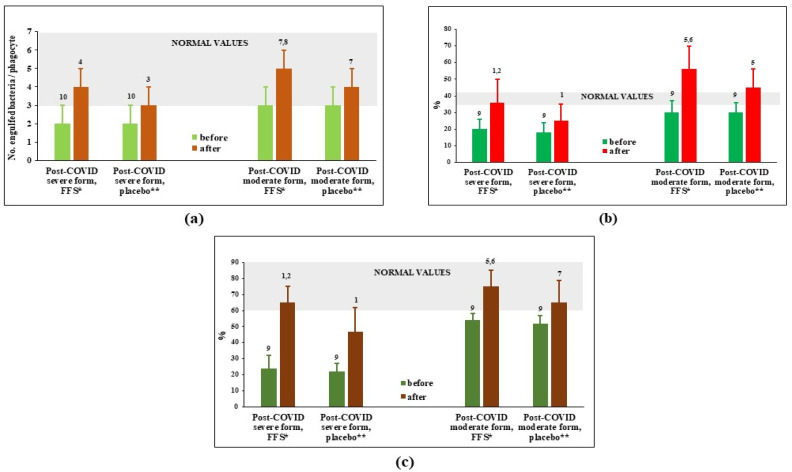
Phagocytosis parameters of circulating polymorphonuclear leukocytes (PMN) before and after the clinical trial. (**a**) Number of engulfed bacteria per single cell; (**b**) actively phagocytosing PMN, phagocytosis index (%); (**c**) intensity of intracellular killing (%). * FFS-Fermented fruit supplement, 28 g daily for 20 days; ** Placebo-diluted 5% honey 28 g daily for 20 days. Grey area covers normal range of values. ^1^
*p* < 0.01 vs. post-COVID severe placebo group before supplementation; ^2^
*p* < 0.01 vs. post-COVID severe FFS group before supplementation; ^3^ 0.05 < *p* < 0.1 vs. post-COVID severe placebo group before supplementation; ^4^ 0.05 < *p* < 0,1 vs. post-COVID severe FFS group before supplementation; ^5^
*p* < 0.01 vs. post-COVID moderate placebo group before supplementation; ^6^
*p* < 0.01 vs. post-COVID moderate FFS group before supplementation; ^7^ 0.05 < *p* < 0.1 vs. post-COVID moderate placebo group; ^8^ 0.05 < *p* < 0.1 vs. post-COVID moderate FFS group before supplementation; ^9^
*p* < 0.01 vs. donors; ^10^ 0.05 < *p* < 0.1 vs. donors.

**Figure 2 nutrients-14-02203-f002:**
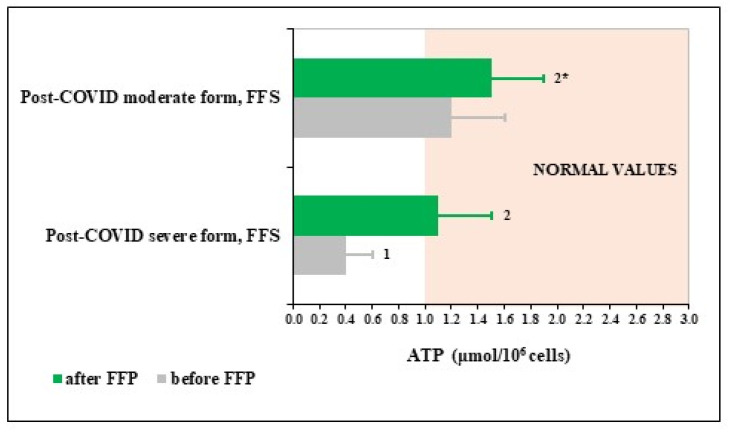
Polymorphonuclear leukocyte (PMN) levels of ATP in post-COVID period of the Fermented Fruits Supplemented (FFS) groups. Area coloured in beige covers normal range of values. ^1^
*p* < 0.01 versus normal values; ^2^
*p* < 0.01 versus post-COVID severe form; ^2^* 0.05 < *p* < 0.1 versus post-COVID moderate form group before FFS supplementation.

**Figure 3 nutrients-14-02203-f003:**
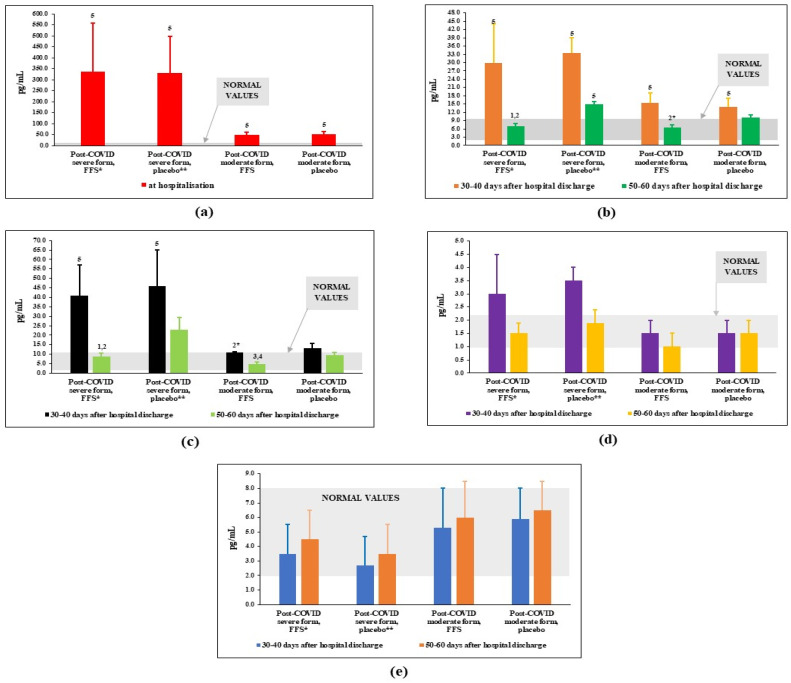
Plasma interleukin levels (pg/mL) in post-COVID period of the FFS and placebo groups. (**a**) IL-6 at the admission to the hospital; (**b**) IL-6 in the post-COVID period before and after supplementation with FFS or placebo; (**c**) IL-8 in the post-COVID period before and after supplementation with FFS or placebo; (**d**) IL-17A in the post-COVID period before and after supplementation with FFS or placebo; (**e**) INF-γ in the post-COVID period before and after supplementation with FFS or placebo. * FFS-Fermented fruit supplement, 28 g for 20 days; ** Placebo-diluted 5% honey 28 g for 20 days. Grey area covers normal range of values. ^1^
*p* < 0.01 vs. post-COVID severe form, FFS; ^2^
*p* < 0.01 vs. post-COVID severe form, 30–40 days after infection (before FFS); ^2^* 0.05 < *p* < 0.1 vs. post-COVID severe form, FFS; ^3^
*p* < 0,01 vs. post-COVID moderate form, placebo; ^4^
*p* < 0.01 vs. post-COVID moderate form, 30–40 days after infection (before FFS); ^5^
*p* < 0.01 vs. donors.

**Figure 4 nutrients-14-02203-f004:**
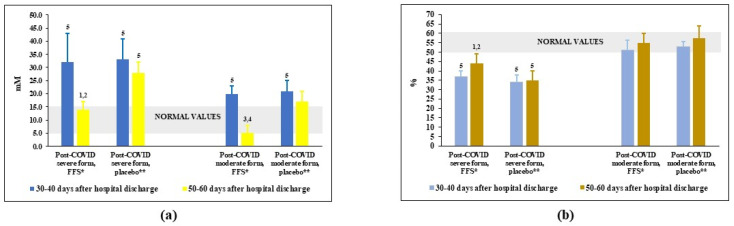
Plasma levels of **(a)** nitrites-nitrates and **(b)** plasma antioxidant capacity (AOA) in post-COVID period of the FFS and placebo groups. * FFS—Fermented fruit supplementation, 28 g for 20 days; ** Placebo-diluted 5% honey, 28 g for 20 days; Grey area covers normal range of values. ^1^
*p* < 0.01 versus post-COVID severe form, FFS; ^2^
*p* < 0,01 versus post-COVID severe form, before FFS supplementation; ^3^
*p* < 0.01 versus post-COVID moderate form, FFS; ^4^
*p* < 0.01 versus post-COVID moderate form, before FFS supplementation; ^5^
*p* < 0.01 versus donors.

**Table 1 nutrients-14-02203-t001:** Demographic distribution of post-COVID patients and donors in the treatment groups.

Group	Number of Patients	Age Range, Years	Sex	
			F	M
Group 1. Post-COVID after severe COVID, fermented fruits supplementation (FFS)	64	38–69	35	29
Group 2. Post-COVID after severe COVID, placebo	27	39–67	13	14
Group 3. Post-COVID after moderate COVID, fermented fruits supplementation (FFS)	68	38–65	32	36
Group 4. Post-COVID after moderate COVID, placebo	29	36–64	15	14
Group 5. Healthy donors, control	25	35–59	12	13

**Table 2 nutrients-14-02203-t002:** Subjective opinions of post-COVID patients (n = 91, severe COVID cases).

Question	Answer, Score	Average Score before the Trial (m ± S.D.), FFS *	Average Score after the Trial (m ± S.D.), FFS	Average Score before the Trial (m ± S.D.), Placebo **	Average Score after the Trial (m ± S.D.), Placebo	Fisher’s Exact Test, Significance (*p*)
Weakness	2—great	2.0±0.0	1.1 ± 0.1	2.0 ± 0.0	1.6 ± 0.1	*p* < 0.05
	1—moderate					
	0—absent					
Impairment of physical working capacity	2—great	2.0 ± 0.0	1.1 ± 0.2	2.0 ± 0.0	1.6 ± 0.1	*p* < 0.05
	1—moderate					
	0—absent					
Impairment of mental working capacity	2—great	2.0 ± 0.0	1.1 ± 0.1	2.0 ± 0.0	1.6 ± 0.2	*p* < 0.05
	1—moderate					
	0—absent					
Memory impairment	2—great	2.0 ± 0.0	1.4 ± 0.3	2.0 ± 0.0	1.6 ± 0.1	0.05 < *p* < 0.1
	1—moderate					
	0—absent					
Attention concentration impairment	2—great	2.0 ± 0.0	1.3 ± 0.3	2.0 ± 0.0	1.6 ± 0.1	0.05 < *p* < 0.1
	1—moderate					
	0—absent					
Headache	2—frequent	1.4 ± 0.1	1.0 ± 0.1	1.4 ± 0.1	1.2 ± 0.1	0.05 < *p* < 0.1
	1—rare					
	0— none					
Dizziness	2—frequente	1.4 ± 0.1	1.0 ± 0.1	1.4 ± 0.1	1.2 ± 0.1	0.05 < *p* < 0.1
	1—rare					
	0—non					
Night sleeping impairment	2—great	1.1 ± 0.1	0.4 ± 0.1	1.2 ± 0.1	0.9 ± 0.1	*p* < 0.05
	1—moderate					
	0—absent					
Sleepiness during day time	2—great	1.1 ± 0.1	0.5 ± 0.1	1.2 ± 0.1	0.9 ± 0.1	*p* < 0.05
	1—moderate					
	0—absent					
Emotional instability (tearfulness, irritation, aggression)	2—great	1.1 ± 0.1	0.5 ± 0.1	1.2 ± 0.1	0.9 ± 0.1	*p* < 0.05
	1—moderate					
	0—absent					
Depression or apathy	2—frequent	1.1 ± 0.1	0.5 ± 0.1	1.2 ± 0.1	0.9 ± 0.1	*p* < 0.05
	1—rare					
	0—none					
Anxiety, suspiciousness	1—Yes	1.2 ± 0.2	0.2 ± 0.1	1.3 ± 0.3	0.3 ± 0.1	*p* > 0.05
	0—No					
Heart pain	2—great	1.1 ± 0.1	0.5 ± 0.1	1.2 ± 0.2	0.9 ± 0.1	*p* < 0.05
	1—moderate					
	0—absent					
Heart arythmia	2—great	1.1 ± 0.1	0.4 ± 0.1	1.2 ± 0.2	1.0 ± 0.2	*p* < 0.05
	1—moderate					
	0—absent					
Tachycardia	2—great	1.1 ± 0.1	0.4 ± 0.1	1.2 ± 0.3	1.0 ± 0.1	*p* < 0.05
	1—moderate					
	0—absent					
Night sweating	1—Yes	0.6 ± 0.1	0.04 ± 0.01	0.6 ± 0.1	0.3 ± 0.1	*p* < 0.05
	0—No					
Joint pain	2—great	1.3 ± 0.3	0.4 ± 0.1	1.2 ± 0.1	0.9 ± 0.1	*p* < 0.05
	1—moderate					
	0—absent					
Muscle pain	2—great	1.3 ± 0.3	0.5 ± 0.1	1.2 ± 0.1	0.9 ± 0.1	*p* < 0.05
	1—moderate					
	0—absent					
Dryness of skin and epithelia	2—great	2.0 ± 0.0	1.2 ± 0.3	2.0 ± 0.0	1.2 ± 0.2	*p* > 0.05
	1—moderate					
	0—absent					
Change of bodyweight	1—increased	1.0 ± 0.0	1.0 ± 0.0	1.0 ± 0.0	1.0 ± 0.0	*p* > 0.05
	1—decreased					
	0—no change					
Hair loss	1—Yes	0.7 ± 0.1	0.7 ± 0.1	0.7 ± 0.1	0.7 ± 0.1	*p* > 0.05
	0—No					
Total average score	-	1.2 ± 0.2	0.5 ± 0.1	1.2 ± 0.2	0.9 ± 0.1	*p* < 0.05

* FFS-fermented fruits supplementation (28 g/day, daily for 20 days); ** Placebo-diluted 5% honey (28 g/day, daily for 20 days); *p* < 0.05 significant difference between groups after the trial; 0.05 < *p* < 0.1 trend to difference between groups after the trial; *p* > 0.05 no difference between groups after the trial.

**Table 3 nutrients-14-02203-t003:** Subjective opinions of post-COVID patients (n = 97, moderate COVID cases).

Question	Answer, Score	Average Score before the Trial (m ± S.D.), FFS * Group	Average Score after the Trial (m ± S.D.), FFS Group	Average Score before the Trial (m ± S.D.), Placebo **	Average Score after the Trial (m ± S.D.), Placebo	Fisher’s Exact Test, Significance (*p*)
Weakness	2—great	1.3 ± 0.3	0.3 ± 0.3	1.3 ± 0.3	0.8 ± 0.5	*p* < 0.05
	1—moderate					
	0—absent					
Impairment of physical working capacity	2—great	0.8 ± 0.5	0.3 ± 0.3	1.0 ± 0.4	0.7 ± 0.2	*p* < 0.05
	1—moderate					
	0—absent					
Impairment of mental working capacity	2—great	1.2 ± 0.4	0.3 ± 0.3	1.1 ± 0.4	0.7 ± 0.2	*p* < 0.05
	1—moderate					
	0—absent					
Memory impairment	2—great	1.1 ± 0.4	0.5 ± 0.3	1.1 ± 0.4	0.9 ± 0.6	0.05 < *p* < 0.1
	1—moderate					
	0—absent					
Attention concentration impairment	2—great	1.1 ± 0.4	0.5 ± 0.5	1.1 ± 0.9	0.9 ± 0.4	0.05 < *p* < 0.1
	1—moderate					
	0—absent					
Headache	2—frequent	0.1 ± 0.1	0.02 ± 0.02	0.2 ± 0.02	0.06 ± 0.06	0.05 < *p* < 0.1
	1—rare					
	0—none					
Dizziness	2—frequent	0.6 ± 0.6	0.1 ± 0.1	0.4 ± 0.1	0.2 ± 0.1	*p* > 0.05
	1—rare					
	0—none					
Night sleeping impairment	2—great	0.5 ± 0.5	0.2 ± 0.2	0.5 ± 0.5	0.3 ± 0.3	*p* > 0.05
	1—moderate					
	0—absent					
Sleepiness during day time	2—great	1.1 ± 0.4	0.6 ± 0.4	0.9 ± 0.9	0.5 ± 0.5	*p* > 0.05
	1—moderate					
	0—absent					
Emotional instability (tearfulness, irritation, aggression)	2—great	1.6 ± 0.4	0.4 ± 0.4	1.8 ± 0.2	0.7 ± 0.6	*p* > 0.05
	1—moderate					
	0—absent					
Depression or apathy	2—frequent	1.6 ± 0.4	0.4 ± 0.4	1.8 ± 0.2	0.7 ± 0.6	*p* > 0.05
	1—rare					
	0—none					
Anxiety, suspiciousness	1—Yes	0.4 ± 0.4	0.1 ± 0.1	0.5 ± 0.5	0.3 ± 0.2	*p* > 0.05
	0—No					
Heart pain	2—great	0.2 ± 0.2	0.02 ± 0.02	0.2 ± 0.2	0.1 ± 0.1	*p* < 0.05
	1—moderate					
	0—absent					
Heart arrhythmia	2—great	1.0 ± 0.2	0.4 ± 0.2	1.1 ± 0.2	1.0 ± 0.2	*p* < 0.05
	1—moderate					
	0—absent					
Tachycardia	2—great	1.0 ± 0.2	0.4 ± 0.2	1.1 ± 0.2	1.0 ± 0.2	*p* < 0.05
	1—moderate					
	0—absent					
Night sweating	1—Yes	0.3 ± 0.2	0	0.3 ± 0.2	0.03 ± 0.03	*p* > 0.05
	0—No					
Joint pain	2—great	0.6 ± 0.2	0.2 ± 0.2	0.6 ± 0.2	0.4 ± 0.2	*p* > 0.05
	1—moderate					
	0—absent					
Muscle pain	2—great	1.0 ± 0.2	0.4 ± 0.2	1.1 ± 0.2	1.0 ± 0.2	*p* < 0.05
	1—moderate					
	0—absent					
Dryness of skin and epithelia	2—great	1.9 ± 0.1	1.0 ± 0.2	1.8 ± 0.2	1.7 ± 0.2	*p* < 0.05
	1—moderate					
	0—absent					
Change of bodyweight	1—increased	0.8 ± 0.2	0.8 ± 0.2	0.8 ± 0.2	0.8 ± 0.2	*p* > 0.05
	1—decreased					
	no change—0					
Hair loss	1—Yes	0.4 ± 0.1	0.3 ± 0.1	0.3 ± 0.1	0.3 ± 0.1	*p* > 0.05
	0—No					
Total average score	-	1.3 ± 0.3	0.3 ± 0.1	1.2 ± 0.2	0.6 ± 0.1	*p* < 0.05

* FFS-fermented fruits supplementation (28 g/day, daily for 20 days); ** Placebo-diluted 5% honey (28 g/day, daily for 20 days); *p* < 0.05 significant difference between groups after the trial; 0.05 < *p* < 0.1 trend to difference between groups after the trial; *p* > 0.05 no difference between groups after the trial.

**Table 4 nutrients-14-02203-t004:** Dynamics of electrocardiography (ECG) in post-COVID patients supplemented with FFS or with placebo: number of patients and per cent of patients having a symptom **after** vs. **before** the trial (%).

Group	Number of Patients	Dysmetabolic Changes in the Myocardium Number of Patients		Cardiac Arrhythmias					
				Partial Blockade of the Left/Right Bundle of Hiss Number of Patients		Bradycardia Number of Patients		Supraventricular Extrasystole Number of Patients	
		30–40 Days	50–60 Days	30–40 Days	50–60 Days	30–40 Days	50–60 Days	30–40 Days	50–60 Days
Post-COVID severe, FFS	64	64	41 (64. 1%) *	11	8 (72.7%) *	8	4 (50.0%) *	21	8 (38.0%)
Post-COVID severe, placebo	27	27	19 (70.4%) *	5	4 (80.0%) *	9	5 (55.5%) *	9	4 (44.4%)
Post-COVID moderate, FFS	68	40	17 (42.5%) *	8	3 (37.5%) *	3	1 (33.3%) *	9	4 (44.4%)
Post-COVID moderate, placebo	29	19	16 (84.2%) *	4	3 (75.0%) *	3	2 (66.6%) *	4	3 (75%)

* Percent of patients having a symptom after vs. before the trial. Several examples of ECG before and after FFS administration are shown in Appendix B, Figure A1.

**Table 5 nutrients-14-02203-t005:** Dyspnea (subjective) and tolerance to physical load *.

Group	Number of Patients	Number of Patients on Respiratory Support before the Trial *	Number of Patients on Respiratory Support after the Trial *	Subjective Dyspnea, Borg’s Score before the Trial	Subjective Dyspnea, Borg’s Score after the Trial	6MWT, Number of Patients with Oxygen Desaturation > 4%, (%) before the Trial	6MWT, Number of Patients with Oxygen Desaturation > 4%, (%) after the Trial
Group 1, severe COVID, post-COVID, FFS *	64	21	6 (28.5%) **	7.1 ± 1.1	9.4 ± 1.9 ^1^	15	8 (53.3%) **
Group 2, severe COVID, post-COVID, placebo	27	9	4 (44.4%) **	7.3 ± 0.8	9.1 ± 1.4 ^1^	7	5 (71.4%) **
Group 3, moderate COVID, post-COVID, FFS	68	0	0	10.8 ± 1.3	11.8 ± 1.3 ^1,2^	ND	ND
Group 4, moderate COVID, post-COVID, placebo	29	0	0	10.5 ± 1.0	10.8 ± 1.3	ND	ND

* Before the trial (30–40 days after discharge from the hospital). Physical load was not applied to the patients of Groups 1 and 2 on respiratory/oxygen support; ** Per cent of patients after the trial vs. before the trial. ^1^—*p* < 0.05 vs. the same group before the trial; ^2^—*p* < 0.05 vs. placebo Group 4.

## Data Availability

Not applicable.

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
