# Peer review of "Fermented Carica papaya and Morinda citrifolia as Perspective Food Supplements for the Treatment of Post-COVID Symptoms: Randomized Placebo-Controlled Clinical Laboratory Study"

_nutrients, 2022, doi:10.3390/nu14112203_

Round 1
Reviewer 1 Report
The manuscript submitted to Nutrients is an important paper that addresses a tandomised placebo-clinical/laboratory study to evaluate the post Covid 19 symtoms using fermented products (Carica Papaya and Morinda citrifolia. The manyscript is of interest given that relates the pandemic with foods/food supplementation. In my opinion, numerous research papers will be conducted the next years on this topic. Therefore, at this time the paper is an original article in nature with novel insights. The manuscript is well organized and written in good English. However, there are some technical errors that authors should correct through the revision process. I have indicated them within the attached pdf.
Based on these comments, I suggest a minor revision prior to further consideration for publication.

Author Response
- We are very grateful to the reviewer for a general positive consideration
- Unfortunate technical mistakes noticed by the reviewer (lines 53-156, line 599, and line 629) were eliminated/corrected (in yellow colour)
Reviewer 2 Report
The paper intiteled "Fermented Carica papaya and Morinda citrifolia as Perspective Food Supplements for the Treatment of Post-COVID Symptoms: Randomised Placebo-Controlled Clinical Laboratory Study" and authored by Zaira Kharaeva , Albina Shokarova , Zalina Shomakhova , Galina Ibragimova , Pavel Trakhtman , Ilya Trakhtman , Jessie Chung , Wolfgang Mayer , Chiara De Luca , Liudmila Korkina fity well the scope of the journal. nutrients-1735888 targets a hot topic that is well discussed by the scientific community. The manuscript design is appropriate and experiments seemy to be conducted rigourously. I really appreciated reading this manuscript. However in my point of view the manuscript can be nicely impèroved in the following sections:
- Introduction : Please explain the rational of using fermented carica papaya and Morinda citrofolia in the treatment of post-covid symptoms. It will highlight the research driven hypothesis concept. Citing all proven effects of the two food supplements used is suitable but please highlight more the reasons of using them for treating post-covid symptoms.
- Results section : the resolution of the figures should be improved. Some figures such as figure 3 are difficult to read.
- Discussion section: I regret that not much studies have been involved in this section. Please enrich this section using relevant studies.
- Conclusion : please highlight more widely the impact of your study. I really find the results so relevant and useful to the scientific community but they are not well exposed.
Please proceed with requested modifications and i will be happy to reconsider the revised version
Best regards
Author Response
- The authors appreciate very much professional, flattering, and friendly opinion on our work and his/her consideration to review a revised version of our manuscript.
- The Introduction was edited in accord with the reviewer’s suggestions (in yellow colour).
- The quality of Figures will be surely improved upon acceptance, if it happens, of the manuscript for publication. The Instructions for Authors speak clearly about this option.
- The Discussion section was edited accordingly as we added current literature relevant to the topic (17-22).
- The Conclusions were edited in order to highlight the impact of the results obtained as per reviewer’s suggestion
Round 2
Reviewer 2 Report
I fully agree with the authors the manuscript have been edited according to my recommandations and I suggest acceptance without further delay.